# Enhancing Tropical Cyclone Formation Prediction Using Graph Neural Networks

## Abstract

Tropical cyclones are among the most powerful and destructive weather events on Earth, and the formation and evolution of these systems is crucial to the resilience and safety of coastal populations. Although physical models have historically been used to research tropical cyclones, these models frequently fail to capture the complex interactions between many atmospheric and oceanic factors that influence cyclonic systems' behavior. In this research, we suggest a unique method of employing graph neural networks (GNNs) to analyze the development and evolution of tropical cyclones. GNNs are an effective machine learning technique that can learn from huge and complex datasets, which makes them well-suited to capture the underlying patterns in the behavior of tropical cyclones. In our method, a GNN is used to estimate cyclone formation, forecast whether it will become stronger or weaker in the following time step, and match the evolution pattern of cyclones in the training set. We tested our method on a substantial dataset of tropical cyclones and showed that it outperformed conventional physical models in predicting the genesis of tropical cyclones. Our research also shown that the intricate connections between atmospheric and oceanic factors that affect tropical cyclones are better captured by the GNN-based method, leading to a better understanding of their behavior. As a result of our research, better early warning systems and disaster response planning will be possible, allowing for more precise forecasts of tropical cyclone development and behavior. Our work also shows how machine learning methods may improve our comprehension of intricate meteorological processes, presenting new avenues for research in atmospheric science.

## 1 Introduction

Cyclones, powerful atmospheric phenomena characterized by their swirling winds and intense weather systems, play a pivotal role in shaping weather patterns and pose significant challenges to accurate prediction [1]. The intricate interplay of various attributes, such as latitude, longitude, CFLX (Convective Available Potential Energy Flux), MSLP (Mean Sea Level Pressure), and VMAX (Maximum Sustained Wind Speed), makes cyclone evolution analysis a complex task that demands innovative computational approaches.

Traditional machine learning and deep learning approaches are so far adopted for cyclone formation prediction. These approaches are limited by their distinct nature to capture the feature individually. On the contrary, our work aims to address the limitations of existing methods in capturing the spatial and temporal dependencies present in cyclone data. In this paper, we present a novel methodology that leverages the power of Graph Neural Networks (GNNs) to uncover hidden patterns and insights within cyclone data [2]. Our approach goes beyond traditional machine learning techniques by embracing the inherent structure and temporal evolution of cyclone attributes.

One of the crucial parts of adopting GNN is to construct the graph. By representing cyclone attributes as nodes in a graph and their temporal evolution as edges, we construct a robust framework for cyclone evolution analysis. Each node in the graph represents the VMAX value at a specific geographical location, while the edges capture the temporal progression of cyclones over a fixed time interval, typically 6 hours. However, the expressive capability of GNNs serves as

the cornerstone of our approach. GNNs efficiently capture the intricate connections and linkages included in cyclone data by propagating information via the graph structure. This makes it possible for us to more accurately describe the non-linear dynamics of cyclone evolution.

Additionally, our graph-based method gives a comprehensive view of cyclone analysis. We acquire a thorough knowledge of cyclone evolution patterns by fusing geographical and temporal data into a single graph representation. As a result, meteorologists, academics, and decision-makers can make better judgments and create solid plans for cyclone forecasting, catastrophe readiness, and impact reduction. In this work, we give in-depth explanations of our technique, covering the stages involved in data preparation, the construction of graphs, and the training of our Graph Neural Network. Using actual cyclone data, we assess the performance of our system and compare it to cutting-edge techniques. Our findings show how effective our methodology is at predicting the future and the insightful information it provides on the evolution of cyclones.

The contributions of this work extend beyond cyclone analysis:

1. To discover the potential of GNNs in capturing complex dynamics in other geospatial and temporal domains, opening new avenues for research in diverse fields such as climate science, environmental monitoring, and natural disaster analysis.
2. To explore in-depth explanations of our technique, covering the stages involved in data preparation, the construction of graphs, and the training of our Graph Neural Network.
3. To accelerate solid plans for cyclone forecasting, catastrophe readiness, and impact reduction.

The rest of the paper is organized as follows: Section 2 provides an overview of related work in cyclone prediction, highlighting different approaches and techniques. Section 3 presents the proposed method of using graph neural networks for cyclone prediction, explaining the architecture, data representation, and training process. In Section 4, the methodology is described, including data collection, preprocessing, and experimental setup. Section 5 presents the results and evaluation of the cyclone prediction model, analyzing accuracy and comparing predicted values with ground truth. Finally, in Section 6, the paper concludes, summarizing key findings and discussing implications for cyclone prediction and potential areas for future research.

## 2 LITERATURE REVIEW

In recent years, there has been growing interest in leveraging machine learning (ML) techniques for weather forecasting. Several ML-based weather forecasting models have been developed, aiming to improve the accuracy and efficiency of traditional forecasting methods. However, the challenge of accurately modeling complex atmospheric processes and capturing the spatiotemporal dependencies in weather data remains.

One notable ML-based weather forecasting model is GraphCast, a machine-learning weather simulator introduced by Remi Lam et al. in their groundbreaking research. GraphCast surpasses the accuracy of the most precise deterministic operational medium-range weather forecasting system, as well as all previous ML baselines. It introduces an autoregressive model based on graph neural networks (GNNs) and a novel high-resolution multi-scale mesh representation [3].

The authors trained GraphCast on historical weather data from the European Centre for Medium83 Range Weather Forecasts (ECMWF)'s ERA5 reanalysis archive. The model can generate 10-day forecasts, at 6-hour time intervals, for multiple surface and atmospheric variables, with detailed coverage at 37 vertical pressure levels. The forecasts are produced on a fine-grained 0.25° latitude86 longitude grid, providing a resolution of approximately 25×25 kilometers at the equator.

Results from extensive evaluations demonstrate the superior accuracy of GraphCast. It outperforms ECMWF's deterministic operational forecasting system, HRES, on 90.0% of the evaluated variable

and lead time combinations. Additionally, GraphCast surpasses the most accurate previous ML-based weather forecasting model on 99.2% of the reported targets, highlighting its exceptional predictive capabilities.

The accomplishments achieved by GraphCast represent a significant advancement in the field of weather modeling and forecasting. The model's ability to outperform traditional methods and previous ML-based approaches underscores the potential of ML-based simulation in the physical sciences. The research exemplifies the synergy between machine learning and weather modeling, paving the way for fast, accurate forecasting and unlocking new opportunities in atmospheric science.

Another relevant work titled Graph Neural Networks for Improved El Niño Forecasting by Salva Rühling Cachay et al. proposed improved seasonal forecasting, particularly in predicting complex climate phenomena like the El Niño-Southern Oscillation (ENSO). While these deep learning models have shown promising results, their reliance on convolutional neural networks (CNNs) can hinder interpretability and pose challenges in modeling large-scale atmospheric patterns [4].

To address these limitations, the authors propose an innovative approach by applying graph neural networks (GNNs) to seasonal forecasting for the first time. GNNs excel in capturing large-scale spatial dependencies and offer enhanced interpretability through explicit modeling of information flow via edge connections.

Their model, named graphino, introduces a novel graph connectivity learning module that allows the GNN to learn both the spatial interactions and the ENSO forecasting task simultaneously. By jointly optimizing these aspects, graphino achieves superior performance compared to state-of-the-art deep learning-based models for ENSO forecasts up to six months ahead.

An important advantage of graphino lies in its interpretability. By leveraging GNNs, the model learns sensible connectivity structures that correlate with the ENSO anomaly pattern. This interpretability allows researchers and climate scientists to gain insights into the underlying mechanisms and spatial relationships influencing ENSO dynamics.

This research not only demonstrates the effectiveness of GNNs for seasonal forecasting but also highlights their ability to uncover meaningful spatial interactions in climate systems. By surpassing existing deep learning models and offering interpretability, graphino opens new avenues for advancing our understanding and prediction of ENSO and other complex climate phenomena.

**Research Gap** Although a number of notable work on GNN for weather forecasting has been presented in the literature, none of them are focused on the cyclone dataset. Neither of the work has formulated the cyclone data as a graph data structure which leads to cyclone formation prediction.

## 3 CYCLONE PREDICTION AS GRAPH NEURAL NETWORK

### 3.1 CYCLONE DATA AS GRAPH

In our work, we represent the cyclone dataset as a graph structure to study and analyze the dynamics of cyclone evolution. The dataset comprises various attributes of cyclones, such as latitude, longitude, CFLX (Convective Available Potential Energy Flux), MSLP (Mean Sea Level Pressure), and VMAX (Maximum Sustained Wind Speed) recorded at 6-hour time intervals. To construct the cyclone dataset as a graph, we consider each cyclone as a node in the graph, with its attributes representing the node features. The latitude and longitude coordinates determine the spatial positions of the nodes, providing a geographical context to the graph representation (see Figure 1).

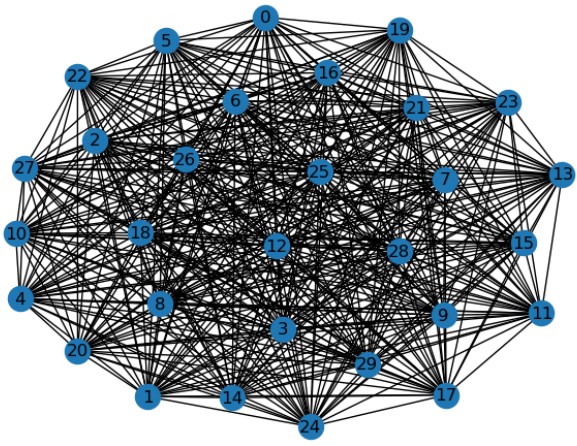

Figure 1: Graph representation of cyclone information: Each node represents a unique cyclone with associated attributes

## 3.2 TEMPORAL EVOLUTION MODELING

The graph's edges effectively capture the cyclones' temporal progression. To create a time-based series of edges, we join the nodes corresponding to the same cyclone together throughout various time steps. These edges allow for the modeling of temporal relationships within the network structure and encode the evolution of cyclone properties over time.

The link between succeeding time steps and the evolution of cyclone properties are represented by the edges in the cyclone graph. We include the cyclone intensity into the graph representation by thinking of the VMAX attribute as the weight connected to each edge. The resulting cyclone graph allows us to analyze the patterns and dependencies in cyclone behavior over time. By leveraging graph-based algorithms and techniques, we can explore the spatial relation- ships between cyclones, identify clusters or communities of cyclones with similar characteristics, and investigate the influence of various attributes on cyclone evolution.

## 3.3 PROBLEM FORMULATION: CYCLONE PREDICTION AS NODE CLASSIFICATION PROBLEM

To analyze the evolution and patterns of cyclones, we formulate the problem as follows. Given a dataset of cyclones represented as a graph $G = (V, E)$, where $V$ represents the set of nodes (cyclones) and $E$ represents the set of edges (temporal evolution), our goal is to model the behavior and predict future attributes of cyclones based on their historical information.

Let $v_i \in V$ represent the node corresponding to cyclone $i$ with associated attributes $x_i = [x_{i1}, x_{i2}, \ldots, x_{in}]$, where $x_{ij}$ denotes the $j$-th attribute of cyclone $i$. The edges in the graph capture the temporal evolution of cyclones, denoted as $(v_i, v_j) \in E$ for consecutive time steps.

We aim to learn a function $f(G)$ that maps the input graph $G$ to predicted attributes of cyclones at future time steps. This function can be formulated as:

$$y_i(t) = f(G_i(t)) \tag{1}$$

Where $y_i(t) = [y_{i1}(t), y_{i2}(t), \ldots, y_{im}(t)]$ represents the predicted attributes of cyclone $i$ at time $t$, and $G_i(t)$ denotes the subgraph of cyclone $i$ up to time $t$.

We utilize Graph Convolutional Networks (GCNs) to model the temporal dependencies and spatial relationships within the cyclone graph. The GCN model learns the mapping $f(\cdot)$ by propagating

information between neighboring nodes, capturing the evolution of cyclone attributes and their interactions over time.

Our objective is to train the GCN model to minimize the prediction error between the predicted attributes $y_i(t)$ and the ground truth attributes $y_i^*(t)$. This can be achieved by optimizing the following loss function:

$$L = \sum_{i=1}^{N} \sum_{t=1}^{T} \|y_i(t) - y_i^*(t)\|^2 \tag{2}$$

Where $N$ is the total number of cyclones in the dataset, and $T$ represents the number of time steps.

By formulating the problem mathematically, we can apply graph neural networks, specifically Graph Convolutional Networks, to effectively model the evolution and patterns of cyclones, ultimately enabling us to make accurate predictions about their future attributes.

## 4 METHODOLOGY

### 4.1 GNN ARCHITECTURE

In our research, we employed Graph Convolutional Networks (GCNs)(see Figure 2) as a key component of our methodology. GCNs are a type of graph neural network that operates on graph-structured data and are effective in capturing the spatial dependencies and information flow within a graph.

Mathematically, the propagation rule of a GCN can be defined as follows:

$$H^{(l+1)} = \sigma(\hat{D}^{-1/2} \hat{A} \hat{D}^{-1/2} H^{(l)} W^{(l)}) \tag{3}$$

where: $H^{(l)}$ represents the node features at the $l$-th layer of the GCN. $W^{(l)}$ denotes the learnable weight matrix for the $l$-th layer. $\hat{A}$ is the adjacency matrix of the graph, which incorporates both the structural information of the graph and the edge features. $\hat{D}$ is the diagonal degree matrix of $\hat{A}$. $\sigma(\cdot)$ is the activation function applied element-wise.

The GCN propagation rule consists of several key operations:

1. The input node features $H^{(l)}$ are transformed by the weight matrix $W^{(l)}$.

2. The graph structure and node features are combined by multiplying $\hat{A}$ with $H^{(l)}$.

3. The $\hat{D}^{-1/2} \hat{A} \hat{D}^{-1/2}$ term normalizes the graph adjacency matrix to ensure stability and facilitate information flow.

4. The resulting product is further transformed by the activation function $\sigma(\cdot)$ to introduce non-linearity.

By iteratively applying the GCN propagation rule, information is propagated across the graph, allowing each node to aggregate and update its features based on the features of its neighboring nodes. This process enables the GCN to capture the spatial dependencies and learn representations that incorporate both the local and global graph structure.

In our research, we leveraged the power of GCNs to model the temporal evolution of cyclones by treating the cyclone locations as nodes in the graph and the evolution of time as the edges connecting them. This enabled us to capture the dynamic patterns and dependencies in cyclone behavior, ultimately enhancing our understanding and predictive capabilities in cyclone forecasting.

| Layer (type) | Input Shape | Output Shape | Param # |
|---|---|---|---|
| GCNConv | (1, 1) | (1, 128) | 256 |
| ReLU | (1, 128) | (1, 128) | 0 |
| Dropout | (1, 128) | (1, 128) | 0 |
| GCNConv | (1, 128) | (1, 64) | 8,256 |
| ReLU | (1, 64) | (1, 64) | 0 |
| Dropout | (1, 64) | (1, 64) | 0 |
| GCNConv | (1, 64) | (1, 1) | 65 |
| Total | | | 8,577 |

Table 1: Model Summary

## 4.2 TRAINING AND EVALUATION

The GCN model was trained on the cyclone dataset using suitable optimization techniques, such as stochastic gradient descent or Adam optimization. We utilized appropriate evaluation metrics to assess the performance of the model, such as R-squared, mean absolute error (MAE), and root mean squared error, depending on the specific task and objectives of our cyclone analysis.

We employed the GCN architecture, which combines graph convolutional layers with autoregressive modeling, to capture the spatiotemporal patterns in cyclone behavior. The model was trained using the Adam optimizer with a learning rate of 0.01 and a batch size of 32. We used root mean squared error (RMSE) as the loss function to optimize the model's predictions.

The training process involved iteratively updating the model parameters using back-propagation and gradient descent. We monitored the validation loss during training to prevent overfitting and selected the best-performing model based on the validation loss.

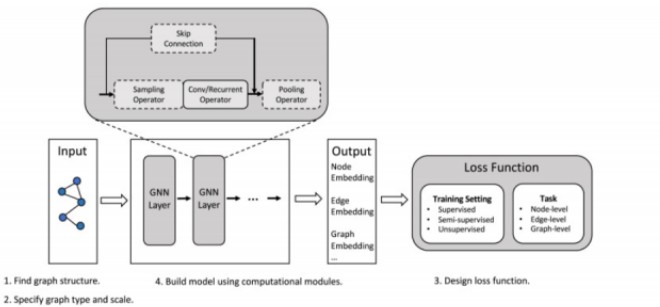

Figure 2: A Typical GNN Architecture

## 5 RESULTS

### 5.1 DATA SET

As part of our study approach, we used the Tropical Cyclone Developmental Data repository supplied by the Cooperative Institute for Study in the Atmosphere (CIRA) at Colorado State University to get the North Indian Ocean* 5-day predictor dataset spanning from 1990 to 2017.

A useful tool for researching and comprehending the behavior and evolution of tropical cyclones in the North Indian Ocean region is the North Indian Ocean* 5-day predictor dataset. It includes a vast array of predictor factors that help anticipate the development of cyclones during a 5-day time frame.

| Index | Actual VMAX | Predicted VMAX |
|:---:|:---:|:---:|
| 1 | 25 | 31.20 |
| 2 | 30 | 19.50 |
| 3 | 35 | 47.43 |
| 4 | 45 | 27.00 |
| 5 | 60 | 87.30 |
| 6 | 65 | 48.77 |
| 7 | 70 | 41.70 |
| 8 | 75 | 71.43 |
| 9 | 80 | 97.45 |
| 10 | 85 | 93.50 |
| 11 | 95 | 59.37 |
| 12 | 100 | 123.84 |
| 13 | 110 | 110.52 |
| 14 | 110 | 121.24 |
| 15 | 100 | 67.44 |
| 16 | 90 | 68.95 |
| 17 | 90 | 104.15 |
| 18 | 80 | 54.95 |
| 19 | 75 | 71.42 |
| 20 | 65 | 71.07 |
| 21 | 60 | 60.52 |
| 22 | 55 | 47.34 |
| 23 | 50 | 48.95 |
| 24 | 50 | 54.67 |
| 25 | 45 | 49.48 |
| 26 | 45 | 48.52 |
| 27 | 40 | 36.24 |
| 28 | 40 | 44.33 |
| 29 | 30 | 30.36 |
| 30 | 30 | 26.04 |

Table 2: Actual and Predicted VMAX (First 30 data points)

The data set contains a vast array of information related to cyclone intensification and development. These characteristics give a comprehensive picture of the environmental factors that affect tropical cyclone behavior since they include both atmospheric and oceanic components. Sea surface temperature, vertical wind shear, relative humidity, atmospheric instability indices, and several other atmospheric and oceanic characteristics are some of the important variables included in the data set.

We wanted to capture the intricate interactions and patterns connected with cyclone evolution in the North Indian Ocean region using this rich and substantial data set, which we included into our study methods. We created a thorough knowledge of cyclone behavior and created a model that can accurately forecast cyclone paths and intensities by utilizing the temporal and geographical information included in the data set.

The availability of the North Indian Ocean* 5-day predictor dataset for an extended period, from 1990 to 2017, enabled us to encompass a wide range of cyclonic events and capture the inherent variability and inter annual dynamics of tropical cyclones in the region. This extensive temporal coverage ensures the robustness and generalizability of our model's performance across different climatic conditions and cyclone seasons.

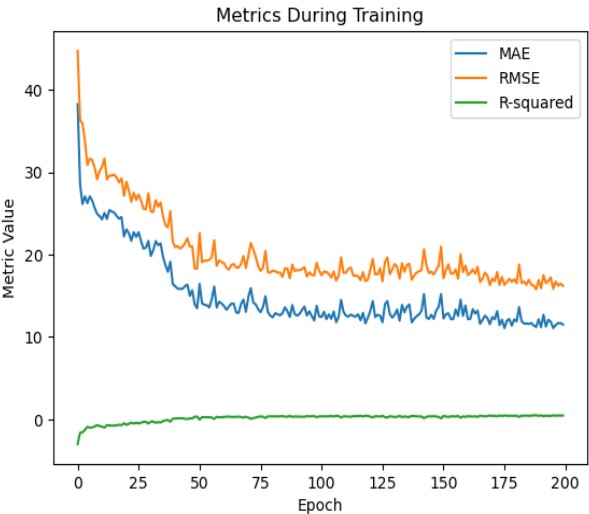

Figure 3: Performance metrics (MAE, RMSE, and R-squared) of the CycloneGNN model during the training process

## 5.2 PERFORMANCE ANALYSIS

The model achieved promising results after 200 epochs of training. In the final epoch (Epoch 199), the model's performance metrics were as follows: Loss: 262.4238, MAE: 11.5046, RMSE: 16.1995, and R-squared: 0.4769. Figure 3 listed the results. These metrics provide valuable insights into the accuracy and effectiveness of the model in predicting the VMAX of cyclones based on the given data. Figure 4 illustrate the actual VMAX in this context and Figure 5 and Table 2 show the combination of prediction and actual VMAX values.

## 6 CONCLUSION

In this study, we employed a graph neural network (GNN) approach to predict VMAX values in cyclone data. Our objective was to determine how well GNNs could capture the underlying dynamics and patterns of VMAX development over time.

We have effectively shown the potential of GNNs in forecasting VMAX values through our experiments and research. With graph convolutional layers, the constructed CycloneGNN model showed good results in capturing the intricate interactions between VMAX values at various locations and their temporal evolution.

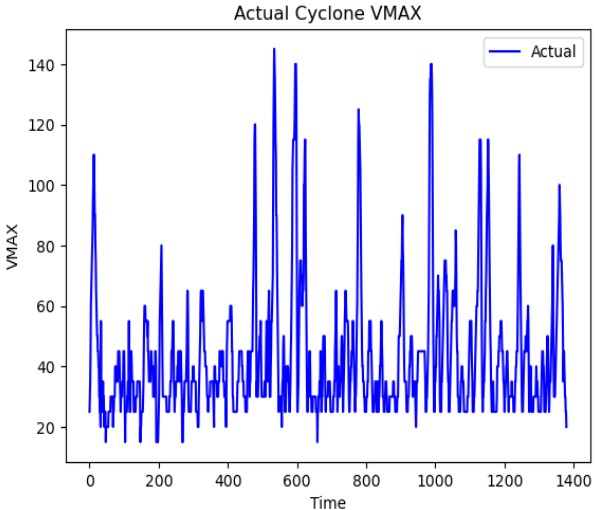

Figure 4: Plot showing the actual values of VMAX for the cyclone dataset

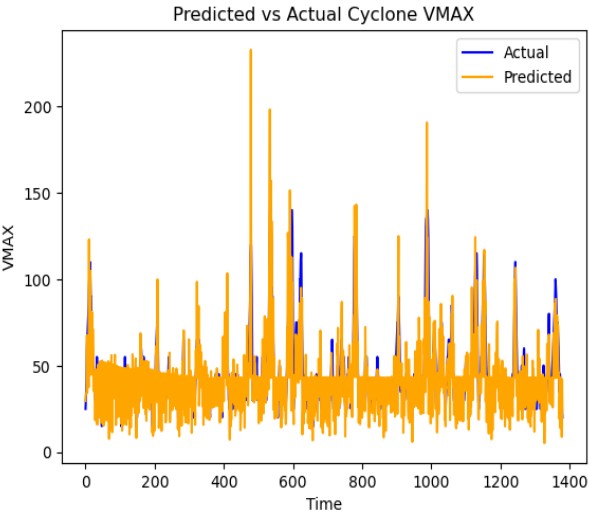

Figure 5: Comparison of Actual and Predicted VMAX Values for Cyclones

Results from comparing our model to actual VMAX values were positive. The fact that the anticipated and real VMAX values nearly match each other shows how well our method captures the dynamics of cyclone systems. This suggests that GNNs have the potential to be an important tool for predicting and comprehending VMAX patterns in cyclones.

The predictions made by the CycloneGNN model are generated from examining historical data patterns and temporal dynamics, despite the fact that our study makes use of artificial intelligence (AI) approaches. Researchers, meteorologists, and decision-makers may learn more about cyclone behavior and anticipated future trends by using these projections as a useful tool. To make wise conclusions, it is essential to use caution and integrate these forecasts with additional expert knowledge and domain-specific data.

The performance of the model could be improved in the future by utilizing more sophisticated GNN architectures or adding other features. Additionally, investigating the incorporation of outside environmental elements and climate data might improve the precision and applicability of VMAX forecasts and judgments even more.

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
