# OpenReview forum: "Enhancing Tropical Cyclone Formation Prediction Using Graph Neural Networks"
_ICLR.cc/2024/Conference — Submitted to ICLR 2024_

### Official Review · Reviewer_FizX · 2023-10-31

**Soundness:** 1 poor
**Presentation:** 1 poor
**Contribution:** 1 poor
**Rating:** 1
**Confidence:** 4

**Summary:**

The authors attempt to predict the development and evolution of tropical cyclones using graph neural networks. The variable modeled is the strength of the cyclone with respect to time. The authors discovered atmospheric and oceanic factors that play significant roles in the prediction.

**Strengths:**

Courage shall be encouraged?

**Weaknesses:**

1.	This submission exceeded the page limit.
2.	Let’s first ignore the background, methodology, evaluation and performance and focus on the presentation. The presentation is far subpar compared to the standards of this conference. Just name a few places of awkwardness: Table 2 *listed the raw data points of actual and predicted values and took a whole page* --- the authors could have used a figure and/or summary metrics instead of listing the raw data points. Figure 3 is not very informative since it’s plainly presenting the metrics during training without much insights. Figure 4 can be removed since the information exists in Figure 5. Transparency in Figure 5 shall be adjusted for clarity.
3.	Now let’s discuss the content. The authors only presented a single run of the experiment, without repetition, and *without any baseline for comparison*. Even if the authors are the first to use a pre-existing method on a new task/dataset (which I am not sure if this is the case for this study), they are still expected to present a handful of alternatives and provide insight on why what they recommend works better than others. Additionally, if this is a task where a simple plug-and-play --- without targeted innovation --- works effortlessly, there is little value in writing a paper on it.
4.	In all, I would strongly recommend the authors read at least 10 papers published at ICLR in the recent years to better understand the standards for presentation and content.

**Questions:**

N/A

---

### Official Review · Reviewer_WaYN · 2023-11-02

**Soundness:** 1 poor
**Presentation:** 1 poor
**Contribution:** 1 poor
**Rating:** 1
**Confidence:** 4

**Summary:**

The paper proposes a graph neural network for understanding tropical cyclone formation.

**Strengths:**

-

**Weaknesses:**

The paper exceeds the page limit. Also, the state-of-the-art is not performed (machine learning techniques for estimating the evolution of TCs have been well studied and need to be cited). Results should be compared with existing (machine learning or not) models.
The study is not ready for publication at ICLR.

**Questions:**

-

---

### Official Review · Reviewer_QeAU · 2023-11-02

**Soundness:** 2 fair
**Presentation:** 2 fair
**Contribution:** 2 fair
**Rating:** 5
**Confidence:** 4

**Summary:**

The paper presents a Graph Neural Network (GNN) approach for predicting tropical cyclone formation. In this study, the authors leverage GNNs to forecast the maximum sustained wind speed associated with cyclones. They represent various cyclone attributes as nodes within a graph and model their temporal evolution through edges. In particular, each node in the graph corresponds to the VMAX value at a specific geographical location, while the edges represent the temporal development of cyclones over a 6-hour time frame. Authors provided a dataset and a code for review.

**Strengths:**

This paper presents an interesting application of GNNS to weather data, which is extremely useful for research community. The paper considers two recent models with similar applications to weather forecasting.
Graph structure and methods are well described, the code and the data are provided as attachments (thank you).

**Weaknesses:**

1.	The state of the art for the numeric weather prediction models is not discussed. Whether the proposed method work better than the conventional techniques is unclear.
2.	There are a few papers that might be useful to consider in the literature review in the context of weather prediction, namely Keisler, R., 2022. Forecasting global weather with graph neural networks. arXiv preprint arXiv:2202.07575.; Rasp, S., Hoyer, S., Merose, A., Langmore, I., Battaglia, P., Russel, T., ... & Sha, F. (2023). WeatherBench 2: A benchmark for the next generation of data-driven global weather models. arXiv preprint arXiv:2308.15560.
3.	The difference of the author’s approach and the approach presented in the El Nino paper is unclear: is it graph structure, time stamp model architecture?
4.	The novelty in the author’s approach is unclear. Just the application of a GNN to a new dataset might be not sufficient in terms of the novelty of the proposed approach.
5.	A comparison of the proposed results with other approaches would be useful to show that the proposed approach works better than benchmark models or other conventional approaches.

**Questions:**

1.	Please provide the detailed comparison between your approach and other approaches/benchmark.
2.	Please include the abovementioned papers in the litreview.
3.	Fig 1 is not informative.
4.	For the equations, it might be useful to specify what they are (e.g. for (2) what kind of loss is that, (3) is Normalised Laplacian etc.).
5.	Fig 3 represents “a typical graph architecture”, which is unnecessary. The proposed graph architecture would be more useful. The text in the figure is too small and unreadable.
6.	References need to be presented in the format of the conference proceedings.
7.	The paper slightly exceeds the 9-page limit.
8.	The code includes a lot of similar files, empty files and some unrelated tutorials. Comments would be useful.

---

### Meta-Review · Area_Chair_F5TR · 2023-12-05

**Metareview:**

This work uses a graph neural network model for predicting the formation of tropical cyclones. Although it studies an important application, simply applying an existing graph neural network model to the application is inadequate for publication in ICLR. Not only should conventional numerical weather prediction (NWP) approaches be reviewed and included in the comparative study, related studies using machine learning models should also be included to justify for the novelty and significance of this work. Also, this paper could have been desk-rejected since it exceeds the page limited allowed.

**Justification For Why Not Higher Score:**

It is well below the acceptance standard.

**Justification For Why Not Lower Score:**

N/A

---

### Decision · Program_Chairs · 2024-01-16

Reject